# Study protocol for 'The Project About Loneliness and Social networks (PALS)': a pragmatic, randomised trial comparing a facilitated social network intervention (Genie) with a wait-list control for lonely and socially isolated people

Rebecca Band, [1,2,3] Sean Ewings, [2] Tara Cheetham-Blake, [2,3] Jaimie Ellis, [2,3] Katie Breheny, [4] Ivaylo Vassilev, [2,3] Mari Carmen Portillo, [2,3] Lucy Yardley, [5,6] Christian Blickem, [7] Rebecca Kandiyali, [8] David Culliford, [3,9] Anne Rogers [2,3]

For numbered affiliations see end of article.

**Correspondence to**
Rebecca Band;
r.j.band@soton.ac.uk

## ABSTRACT

**Introduction** Loneliness and social isolation have been identified as significant public health concerns, but improving relationships and increasing social participation may improve health outcomes and quality of life. The aim of the Project About Loneliness and Social networks (PALS) study is to assess the effectiveness and cost-effectiveness of a guided social network intervention within a community setting among individuals experiencing loneliness and isolation and to understand implementation of Generating Engagement in Network Involvement (Genie) in the context of different organisations.

**Methods and analysis** The PALS trial will be a pragmatic, randomised controlled trial comparing participants receiving the Genie intervention to a wait-list control group. Eligible participants will be recruited from organisations working within a community setting: any adult identified as socially isolated or at-risk of loneliness and living in the community will be eligible. Genie will be delivered by trained facilitators recruited from community organisations. The primary outcome will be the difference in the SF-12 Mental Health composite scale score at 6-month follow-up between the intervention and control group using a mixed effects model (accounting for clustering within facilitators and organisation). Secondary outcomes will be loneliness, social isolation, well-being, physical health and engagement with new activities. The economic evaluation will use a cost-utility approach, and adopt a public sector perspective to include health-related resource use and costs incurred by other public services. Exploratory analysis will use a societal perspective, and explore broader measures of benefit (capability well-being). A qualitative process evaluation will explore organisational and environmental arrangements, as well as stakeholder and participant experiences of the study to understand the factors likely to influence future sustainability, implementation and scalability of using a social network intervention within this context.

**Ethics and dissemination** This study has received NHS ethical approval (REC reference: 18/SC/0245). The

## Strengths and limitations of this study

► This study will evaluate an existing social network intervention (Genie, Generating Engagement in Network Involvement) in the context of loneliness and social isolation.
► The Project About Loneliness and Social networks study consists of a pragmatic randomised controlled trial implemented in conjunction with community-based stakeholders in a community setting in two areas of the UK.
► The process evaluation and analysis has been designed to understand the factors influencing the implementation and scalability of social network interventions in this context.

findings from PALS will be disseminated widely through peer-reviewed publications, conferences and workshops in collaboration with our community partners.
**Trial registration number** ISRCTN19193075

## INTRODUCTION

Social isolation is considered to be an objective lack of social connections, contact or participation, while loneliness is a subjective psychological state where there is a discrepancy between desired and perceived levels of support or connectedness.[1 2] The prevalence rates of loneliness and isolation vary;[3] however, it is estimated to affect about 30% of the adult population in the UK.[4] Specific at-risk groups, such as the elderly, minority communities and those with long-term mental or physical health conditions are significantly more isolated than those in good health.[3 5 6] The Office of National Statistics

(ONS) recently identified three profiles of individuals who are 'at-risk'; these suggest that different factors may be important in the experience of loneliness at different points across the life course.[7]

## The problem: health implications of loneliness and social isolation

The impact of loneliness and isolation on well-being and the associated health risks have been identified as a significant public health concern[8 9] exacerbated by the prevalence of long-term conditions and advancing age.[10] Both loneliness and social isolation are associated with poor physical and mental health outcomes,[11–13] reduced quality of life[14 15] and is linked to poorer physiological outcomes such as raised blood pressure and increased health-risk behaviours (eg, sedentary behaviour).[16] Their impact on mortality is estimated to exceed that of traditional risk factors such as obesity and cigarette smoking, with a 50% higher risk compared with socially integrated participants.[17–19] There are also significant costs associated with raised demand and use of health services, and loneliness is associated with increased general practitioner (GP) appointments, emergency hospital admittance and premature social care use.[20–22]

## Social relationships and preventing or reducing loneliness and social isolation

Although the determinants of loneliness and isolation are varied, social and emotional support from others is likely to be protective,[23] with emerging evidence suggesting that improving the quality of interpersonal relationships and participation in social activities may be key to tackling the impact of loneliness.[9] Evidence has indicated that increasing social interactions and the number of people who can be relied on is associated with reduced levels of distress,[24] while connecting with community resources can help protect against loneliness for those who are most at risk.[9 25] Furthermore, there is evidence that social network interventions can significantly improve health outcomes, quality of life and increase the take-up of new activities.[26 27] A diverse and supportive network has been shown to reduce health service costs.[28] A recent The National Institute for Health and Care Excellence (NICE) quality standard recommends the navigation of older vulnerable people to community activities as a means of preventing loneliness in this group.[25]

## Rationale and risk benefits for the current trial

In line with this evidence, there is a logical argument for introducing an effective social network intervention outside of formal healthcare settings to connect people who are experiencing loneliness to others within their communities.[25] Creative engagement with non-traditional informal providers of wellness management (such as through accessing locally available community groups) offers an alternative opportunity to address health and social needs. We envisage that the study will offset any burden through providing wider benefit to organisations;

first through staff development and training integrating the intervention into practice, and, second, by providing a resource and alternative referral pathway for individuals who they have identified at risk of isolation or loneliness (potentially extending beyond the life of the study). A series of nestled qualitative process studies will examine the context, practices and processes relating to implementing the intervention within the community context, and an economic evaluation to assess whether this is cost-effective.

## Study aims and research questions

The aim of the Project About Loneliness and Social networks (PALS) study is to assess the feasibility, acceptability, effectiveness and cost-effectiveness of a facilitated social network intervention compared with a wait-list control within a community setting among at-risk populations, and to understand the implementation in the context of different organisations who work in this environment. The Genie (Generating Engagement in Network Involvement) intervention is an online, facilitated, social networking tool designed to develop opportunities for social involvement.

Primary objectives
- ▶ To determine the effect of Genie compared with usual care on mental health (SF-12 composite scale score) at 3 and 6 months.

Secondary objectives
- ▶ To determine the effect of Genie compared with usual care on loneliness, social isolation, physical health and engagement with new activities at 3 and 6 months.
- ▶ To establish whether the use of Genie within a community setting is cost-effective.

Process analysis objectives
- ▶ To assess the acceptability and feasibility of running the study based on recruitment and retention during an internal pilot phase.
- ▶ To explore the experiences of using Genie, how the intervention impacts on loneliness and isolation and the mechanisms by which participants enact change.
- ▶ To explore contextual environmental and organisational factors that inhibit or promote the integration, sustainability and scalability of Genie for addressing loneliness in local and organisational settings.

## METHODS AND ANALYSIS
### Study design and setting

We will conduct a pragmatic, randomised controlled trial comparing participants receiving the facilitated social network Genie intervention to a wait-list control group; randomisation will be at individual and/or cluster (facilitator) level (see Randomisation section). We will work closely with community partners two localities (centred around Southampton and Liverpool) in identifying participants and delivering the intervention, as well as informing our understanding of the challenges and environmental factors associated with implementation. Partners may include any group or

organisation that has the potential to identify or access at-risk individuals.

## Study participants
### Identification
We will use a multistranded recruitment strategy to reflect the diversity of individuals who are living with loneliness or in isolation. This will be facilitated by collaborating community organisations to ensure that we are able to identify and access those most at-risk. Potential participants will be identified in the manner that best operates within existing working practices for each organisation (which will be different for each organisation/collaborator). This is necessary to explore the integration and scalability of Genie in local and organisational settings. Potential participants will be invited by the organisation; this may be by letter or during routine visits, appointments, or in line with the usual working practices of the partner organisation. This may include (but is not limited to) new referrals, waiting lists or opportunistic contacts during routine work of partner organisation. All eligible participants will be given a research pack including an invitation letter, participant information sheet and free-post reply slip to return should they wish to take part in the trial.

### Eligibility criteria
We will recruit any adult (aged 18 or over) who is identified as being isolated or at risk of loneliness. We define a socially isolated person as one for whom there is an 'absence of social contacts or community involvement, or lack of access to services' in line with the definition used by Hampshire County Council.[29]

### Exclusion criteria
Exclusion criteria will include participants who are
- currently hospitalised (ie, not self-managing within a community setting),
- ,those in the end stages of life or any condition which impacts on ability to take part,
- those lacking sufficient capacity
- and those having previously used the Genie intervention.

Eligibility will be assessed by the community partners and confirmed by the research team in all cases.

## Randomisation
To overcome potential issues of contamination (where a facilitator could become familiar with how GENIE works and thus find out about local activities and advise control group participants), facilitators will be 1:1 randomised to either the intervention or control arm. Randomisation will be stratified by organisation in blocks of two (ie, one facilitator will be randomised to the intervention arm and one to the control arm) and carried out by the trial statistician (SE) using the statistical software R v3.5.1. In some organisations, the risk of contamination does not exist (eg, where the only contact between a facilitator and participants would be by delivering the intervention). In these cases, some efficiency is gained by randomising the participants individually (block randomised 1:1 to intervention and control, again stratifying by organisation). The summary below details the different scenarios. Preferred options (A&B) to be used whenever possible:
- Randomise facilitators and participants
- Intervention facilitators only to be trained

If there are organisational or setting constraints which prohibit facilitator or participant randomisation (eg, the facilitator works within a specified geographic location) we will assess whether there is ongoing contact between the facilitator and potential participants. In these scenarios:
- Where there is ongoing contact (C)
- Train intervention facilitators only
- Randomise facilitators only
- Participants within each area allocated to facilitator (not randomised)
- Where there is no ongoing contact (D)
- Train all facilitators
- Randomise participants

## Participant flow through the study
Written informed consent will be collected from all participants and baseline data collected with a research team member (online or on paper, dependent on the participant preferences). Allocation will occur once the baseline assessment has been completed. Participants who are allocated to the intervention condition will be given access to the Genie intervention within 2 weeks of the baseline appointment; this process will be guided by

---

**Table 1** The factors affecting the recruitment and randomisation process

| Participant recruitment | Contact between participant and facilitator | |
| --- | --- | --- |
| | Ongoing | One-off contact (at facilitation) |
| Area/location not restricted | MODEL B <br> ► Randomise facilitators and participants <br> ► Intervention facilitators only to be trained | MODEL A <br> ► Randomise facilitators and participants <br> ► Intervention facilitators only to be trained |
| Within a specific geographical (or other prespecified) area | MODEL C <br> ► Train intervention facilitators only <br> ► Randomise facilitators only <br> ► Participants within each area allocated to facilitator (not randomised) | MODEL D <br> ► Train all facilitators <br> ► Randomise participants |

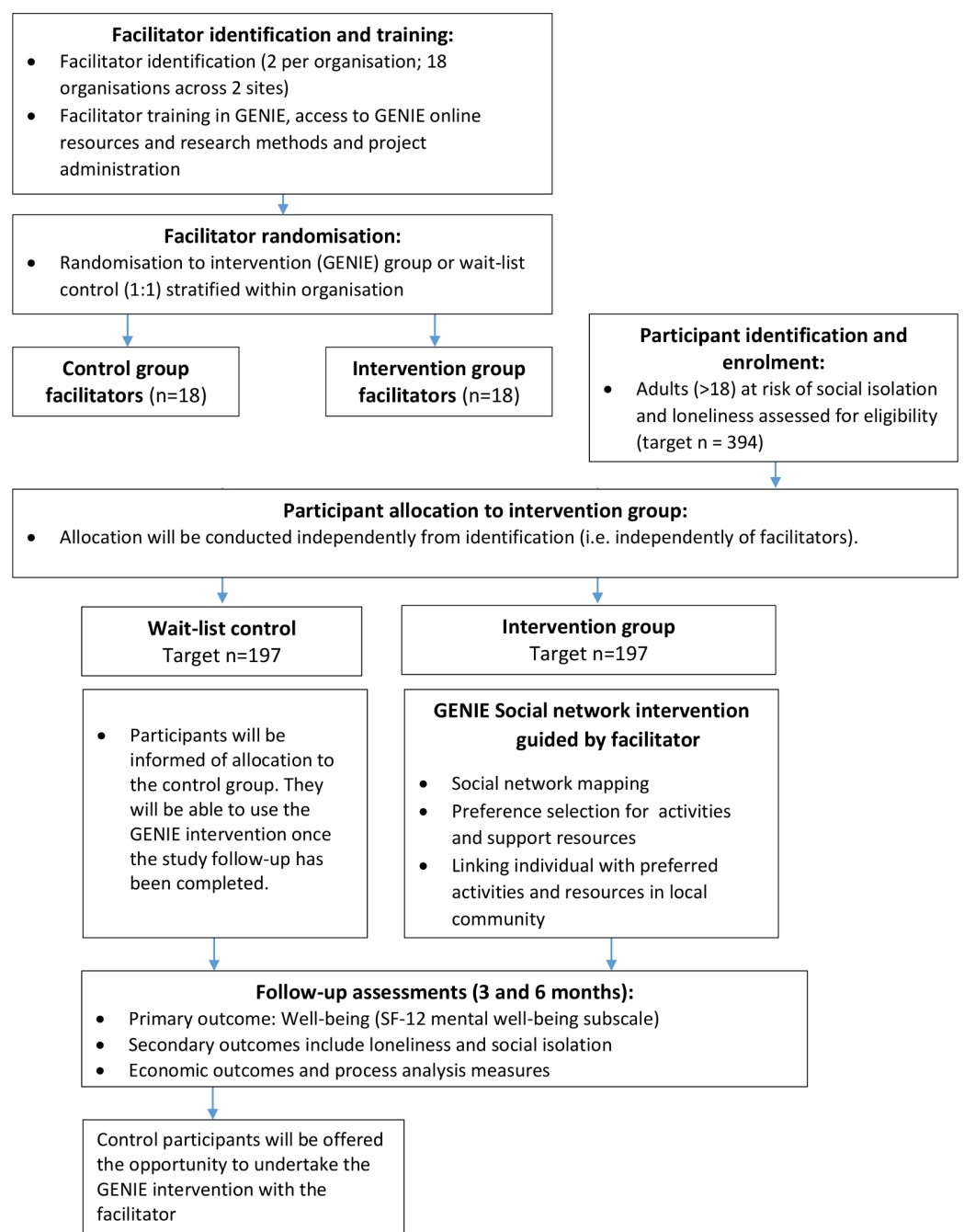

**Figure 1** PALS study flow diagram. GENIE, Generating Engagement in Network Involvement; PALS, Project About Loneliness and Social networks.

the facilitator at a location to suit them (ie, at home or in the community). At 3 and 6 months after enrolment into the study, participants will be invited to complete follow-up assessments. All follow-up assessments will be recorded no earlier than 2 weeks before the follow-up date and no later than 6 weeks after the follow-up date. Each participant will be sent a £10 high street gift voucher with the 6-month follow-up questionnaire. Individuals allocated to the control group will be offered access to Genie with the facilitator after the have completed their

6-month follow-up assessment. Participant flow is outlined in figure 1.

### Sample size consideration

The sample size calculation is based on the primary analysis of the comparison of intervention and control arms on SF-12 Mental Health composite scale (MCS) score at 6 months,[30] and accounts for possible intracluster correlation (ICC) within facilitators. The MCS compares an individual score with an age group mean score; a negative

score reflects poorer health. Previous studies (although in different populations) have suggested that differences of 3 and 4.7 points on the SF-12 would be clinically meaningful.[31 32] We have based the current sample size on being able to detect a difference of 4 points. Based on a previous study in socially isolated older people,[33] we estimate the SD of the outcome to be 10.4 (using a pooled estimate of baseline scores). Choosing 80% power and a type I error rate of 5%, an individually randomised study would require 216 people (108 per arm). Regarding clustering, previous studies have generally shown low ICCs for mental health scores from SF-12 and SF-36 (0.032 and below, although for different populations and clustering within GP practices)[33 34]; we use an ICC of 0.05 here. Based on discussions with participating organisations, it was agreed that 12 participants per facilitator was suitable; this results in a design effect of 1.55 and an adjusted sample size of 335 people. Assuming 15% drop-out,[35] we require 394 participants in total (197 per arm). This requires 33 facilitators; we will increase this to 36 facilitators to account for potential drop-out of facilitators.

## The facilitated social network intervention

The intervention process is introduced initially via a guided discussion with a trained facilitator, takes 30–40 min to deliver and has three stages: social network mapping, tailoring of preferences and linking users to valued resources and activities. By design, Genie (https://pals.genie-net.org/eng/), can be applied to varied user groups.[27] It is based on evidence of social network properties and types, mechanisms and work relating to managing health and wellness.[36–39] Previous testing of the principles has shown that it is both appropriate and acceptable to implement for individuals with a long-term condition.[26–28]

### Facilitators

Guided facilitation is an important element of the guided process to using the tool. Facilitators do not need a specific in-depth theoretical knowledge: instead, the local knowledge of facilitators is important and adds to the value of the intervention. However, the interpersonal skills of the facilitator are vital for the success of engagement through promoting a collaborative solution, and engaging participant focus, motivation and reflection on social network composition and promoting new community engagement.[27] Facilitators receive a minimum of a half-day training from the research team, which may be refreshed over the course of the study. This will include a background to, demonstration of and practical pair-working exercises using video guides around the facilitation process. Research methods training and discussion around loneliness and isolation are also addressed. The research team will provide ongoing support to monitor fidelity to the intervention deployment and address issues arising regarding complex cases (or facilitator difficulties and distress).

### Social network mapping

Facilitators guide participants to create a visual map of their current support network, using a concentric circles method.[27] The concentric circles process provides insight into the user's current situation regarding social support; who they view as important in their daily lives (this may include family members, friends, acquaintances, healthcare professionals, local groups and pets); and then to reflect on renegotiating existing roles and responsibilities, and further map people and groups who could provide extended support.[26–28] This process, when guided by the facilitator, helps the participant to realign thinking about their relationships (and conceptualise themselves within a network of support), explore family dynamics and recognise 'weak ties' (ie, social acquaintances) that already exist in their network.[27] It also offers the opportunity to begin discussions about how support may be extended within the network.

### Linking individuals with preferences and valued local and online activities and resources

The next step involves facilitating access to local resources based on personal preferences, and acceptability to encourage engagement with personal choices, through a set of 13 questions.[40] The questions generate a set of preferred local and online resources (linked to a precreated database of categorised local organisations and resources). The facilitated discussion of preferences is linked to available and accepted potential support from people in a person's network. Personalised results are presented in a user-friendly way aided by Google maps with clear details about access. Previous work has highlighted that this is often new and previously unthought about information for participants.[27] The network maps, description of individual networks, preferences and the local and online resources identified as relevant by individuals can be printed to keep or reaccessed online later via a personalised Genie page.[40 41] Two weeks after the intervention, all Genie users receive a phone call from the facilitator and alternative or additional engagement activities are discussed. The follow-up call takes up to 10–15 min.

### Wait-list control group

All participants allocated to the control group will be offered the opportunity to use the Genie intervention with a facilitator once the 6-month follow-up has been completed to avoid increasing inequalities as a result of the study, particularly for participants living in marginalised and deprived domestic situations.

### Patient and public involvement

Several of our partner organisations were involved in the development of the study and protocol, particularly contributing to understanding methodological issues around identifying participants. We will continue to work closely with all stakeholders in a pragmatic and flexible way to assess implementation issues throughout

the study. Patient and public involvement (PPI) representatives were consulted in the development phase of the study, as well as discussion with the CLAHRC Wessex Wiserd group, and prior Genie engagement work. In addition, further PPI representatives have been included in the trial management group, and we have consulted with the user-led McPin organisation, who are represented on our Steering committee. We will involve our PPI representatives in the interpretation of the findings from our studies, particularly those of user views.

## Outcomes

The primary outcome of the trial will be the SF-12 MCS score at 6-month follow-up between the intervention and control group using a mixed-effects model (accounting for clustering within facilitators and organisation).

Secondary outcomes will include
- SF-12 MCS score between the intervention and control group at 3-month follow-up.
- SF-12 Physical Health composite score between intervention and control groups at 6-month follow-up.
- Loneliness between intervention and control groups at 3-month and 6-month follow-up measured using the De Jong Loneliness scale[42] and the Campaign against loneliness measure.[43]
- Social isolation between intervention and control groups at 3-month and 6-month follow-up measured using the Duke Social Support index.[44]
- Well-being measured using Warwick Edinburgh Mental Well-being scale.[45]
- Participant engagement with new activities.

Economic evaluation measures will include
- Quality-adjusted life year (QALYs) (incremental QALYs) between intervention and control at 6 months, with health-related quality of life calculated using the SF-6D utility algorithm (derived from SF-12 data).[46]
- Incremental costs of public sector resource between intervention and control at 6 months.
- Cost utility (expressed in terms of cost/QALY and cost/year of sufficient capability).
- Capability well-being measured using the ICEpop CAPability Measure for Adults (ICECAP-A) scores between intervention and control at 6 months.[47]

Process evaluation measures will include
- Participant perceived collective efficacy measured using the Collective Efficacy in Networks Scale[48] and social support using the Social Provisions Scale (SPA).[49]
- Perceptions of loneliness measured using a modified version of the Brief Illness Perception questionnaire.[50]

Intervention group only
- Social network composition change measured using Genie social network mapping (intervention group only at 3 months).

**Table 2** Measures and schedule of observations within the PALS study

| Measure | Time point (month) | | |
| --- | --- | --- | --- |
| | Baseline | 3-month follow-up | 6-month follow-up |
| Sociodemographic measures | X | | |
| Patient self-report measures (both groups) | | | |
| SF-12 Mental Health | X | X | X |
| SF-12 Physical Health | X | X | X |
| Loneliness (De Jong Scale) | X | X | X |
| Social isolation (Duke Social Support index) | X | X | X |
| Campaign to End Loneliness scale | X | X | X |
| Collective Efficacy in Networks Scale (CENS) | X | X | X |
| Social support (SPA) | X | X | X |
| Warwick Edinburgh Mental Well-Being Scale (SWEMWBS) | X | X | X |
| Perceptions of loneliness (modified B-IPQ) | X | X | X |
| Participant engagement with new activities | X | X | X |
| Patient measures (network mapping, intervention group only) | | | |
| Social network composition change | X | X | |
| Economic measures | | | |
| SF-6D | X | X | X |
| Capability well-being (ICECAP-A) | X | X | X |
| Health and social care use | X | X | X |
| Process evaluation | | | |
| Qualitative interviews with participants | X | X | X |
| Qualitative interviews with facilitators and stakeholders | X | X | X |
| Observations of facilitation | X | | |
| Community staff observations of impact | X | X | X |

PALS, Project About Loneliness and Social networks.

## Study endpoints

At 3 and 6 months after enrolment in to the study, patients will be invited to complete follow-up assessments. They may do this independently or with the assistance of the facilitator or a research team member (which may include online, on paper or over the phone). All follow-up assessments will be recorded no earlier than 2 weeks before the follow-up date and no later than 6 weeks after the follow-up date.

## Measures

See table 2 for full details of study measures.

## Statistical analysis

All analyses will emphasise estimation and CIs over hypothesis testing, and will be conducted as intention-to-treat.

Missing data will be assumed to be missing at random, unless accounting for more than 10% of the sample; if missingness is above this rate, approaches for dealing with missing data (eg, multiple imputation) will be discussed within the research team. Missingness will be reported for each arm and summaries of baseline characteristics of those lost to follow-up and those not will be used to judge potential sources of bias.

Baseline sociodemographic data will be summarised within randomised arms using appropriate descriptive measures; likewise, all outcome measures will be summarised by arm at each timepoint. We will produce a forest plot of estimated effects for each outcome within each organisation to explore any variability in the impact of the intervention.

The primary analysis will involve a mixed-effects model (pending the model meeting the associated assumptions) comparing groups on SF-12 at 6 months. The model will include a random intercept for facilitator and organisation, with participants clustered within facilitators clustered within organisation (hence a three-level model) and control for baseline SF-12. This analysis will be complemented by an analysis using the same framework but with SF-12 as the outcome and a random coefficient for time, where repeated measurements are clustered within participants (hence a four-level model).

Non-response bias (ie, where a particular group of participants are unavailable or refuse to participate) will be reduced by taking steps to increase the initial response rate and reduce drop-out over the course of the study.

### Economic analysis

The primary analysis will be a cost-utility analysis from a public sector perspective, with a primary outcome of cost/QALY at 6 months. Health-related quality of life will be collected via SF-12 at baseline, 3 and 6 months, with utilities being derived by application of the SF-6D scoring algorithm.[46] In addition, scored values from the capability well-being measure (ICECAP-A)[47 51] will enable a secondary cost-utility analysis.[47] The use of ICECAP–A is planned to explore non-health attributes (specifically capabilities) that might be important to this population, thus allowing for a broader measurement of well-being than might be captured by SF-6D. While the comparative data collected on both measures may inform future studies in similar populations, it will also provide decision-makers with richer information than would be obtained by a single generic Health realted Quality of Life (HRQoL) measure.

Intervention delivery resource use will be recorded on proformas designed to capture cost categories (eg, trainer time, pay scale, intervention setting and facilitator travel costs). Additionally, at baseline, 3 and 6 months resource use will be collected directly from participants using a questionnaire designed to capture healthcare, social service and other public sector service use, as well as participant service use (ie, participant and carer costs). An exploratory analysis will use a societal perspective

providing decision-makers with evidence to inform judgements on what, in the broadest sense, is optimal for society.[52] The analysis of costs from a societal perspective will therefore provide detail on the cost-shifts within sectors (eg, health compared with social care). All analyses will follow practice guidelines,[53–55] including those related to public health and/or complex interventions specifically.[56–58] Cost-utility analysis will also allow for the construction of cost-effectiveness acceptability curves to assess whether the intervention is cost-effective at a range of payer thresholds.[59] Subgroup analysis will be carried out in order to inform policy-makers' decision-making with respect to the targeting of the intervention. Such subgroup analyses (for instance, looking at intervention effects in different groups) will be planned prospectively, and quantitative analysis —foreseeably including mixed-effects modelling to account for the clustered nature of the data[60] —will be set out as part of the statistical and health economic analysis plan. The economic evaluation will also be informed by the process evaluation in terms of considering how the contexts of this complex intervention relate to resource use and cost areas.[61] Such an explanatory focus will be taken throughout the study, with a view to interpreting study results and assessing study generalisability.

### Qualitative process evaluation and analysis

The qualitative process evaluation will combine complementary components to seek to provide an in-depth understanding of the factors that facilitate individual, environmental and organisational factors that inhibit or promote the engagement, workability, integration, sustainability and scalability of a social network intervention for addressing loneliness in open settings. The process evaluation will consider the preimplementation contexts and processes, as well as observing use of the intervention in practice to understand the dynamics of implementation (including how the facilitation and other elements work) to consider implications for scale-up and sustainability for the participating organisations. Concepts from the Consolidated Framework for Implementation Research[62] will be used to guide the identification of factors promoting or inhibiting the routine incorporation and embeddedness of a facilitated social network intervention. The non-adoption, abandonment and challenges to the scale-up, spread and sustainability framework will inform the evaluation of implementation because it has been designed to help predict and evaluate the success of a technology-supported health programme, addressing concerns such as implementation, scale-up and sustainability.[63] An ethnographic approach making use of observations, interviews and documentary analysis will be used to capture the preimplementation processes in order to explore the workability and integration of Genie in different community organisations. Following this, interviews will take place to explore engagement, sustainability and scalability. Participants will be sampled purposively based on description of circumstances of loneliness and

sociodemographic factors (age, gender and locality); we will explore the experiences and meaning of loneliness with reference to social and personal circumstances (eg, living and working arrangements) and situational contexts of loneliness (such as migration, separation and unemployment). This will be combined with exploration of how individual circumstances shape engagement with different elements of the intervention, how change is enacted and embedded into people's everyday lives and how this involves other members of a person's network. We will describe the engagement and activities undertaken following the intervention including how links with new networks and resources are identified and made (navigation); how these are integrated (negotiation) and how new connections improve capacity to enact healthy behaviours, improve well-being or reduce isolation (collective efficacy). We will explore how facilitators felt about delivering Genie and how this might be adopted by their organisations as part of their practice. We will draw out new improvements and benefits specific to individual circumstances and existing use of healthcare services. Further interviews postintervention will be conducted until 'saturation' (ie, no significant new insights emerge).

## ETHICS AND DISSEMINATION
### Data monitoring

The Programme Steering Committee is responsible for ensuring programme adherence to the protocol, and adherence to the requirements of the Guidelines for Good Clinical Practice. The trial may be subject to inspection and audit by University of Southampton, under their remit as sponsor, the trial coordinating centre as the Sponsor's delegate and other regulatory bodies.

### Dissemination

The findings from PALS will be disseminated widely through peer-reviewed publications, scientific conferences and workshops. In addition, we will aim to disseminate through multiple community pathways in collaboration with our partners and stakeholders (including local councils, NHS trusts and other local and national organisations) through interactive methods, such as targeted workshops, podcasts or blogs. If successful, we aim to produce a user guide for applying Genie to loneliness and isolation.

**Author affiliations**
[1]Psychology, University of Southampton, Southampton, UK
[2]Health Sciences, University of Southampton, Southampton, UK
[3]NIHR Collaboration for Leadership in Applied Health Research and Care Wessex, NIHR, Wessex, UK
[4]Population Health Sciences, University of Bristol, Bristol, UK
[5]Academic Unit of Psychology, University of Southampton, Southampton, UK
[6]School of Psychological Science, University of Bristol, Bristol, UK
[7]Public Health, Liverpool John Moores University, Liverpool, UK
[8]Centre for Child and Adolescent Health, University of Bristol School of Social and Community Medicine, Bristol, UK
[9]Faculty of Medicine, University of Southampton, Southampton, UK

**Contributors** RB developed the initial idea for the study and obtained funding in collaboration with AR, SE, DC, IV, MCP, LY, RK, CB and the PALS study team. All authors have contributed to the protocol development. RB, TCB, JE and AR have led the trial preparations and development of training materials. IV has lead the development and modification of Genie. SE and DC led the statistical analysis planning, and KB and RK have led the health economics planning. JE, CB and AR have led the qualitative process evaluation planning. RB wrote the initial draft, all subsequent drafts were contributed to by all authors who have approved the final version.

**Funding** This protocol paper summarises independent research funded by the National Institute for Health Research (NIHR) under its Public Health Research programme (Grant Reference Number 16/08/41). The views expressed are those of the author(s) and not necessarily those of the NHS, the NIHR or the Department of Health and Care.

**Competing interests** None declared.

**Patient consent for publication** Not required.

**Ethics approval** Ethical approval for the PALS study has been obtained from the South Central—Berkshire B ethics committee (reference: 15/SC/0245). All substantial amendments must be approved by the University ethics committee and NHS ethics committee responsible for the trial, in additional to approval by HRA. Investigators are kept up to date with relevant changes via regular management group meetings.

**Provenance and peer review** Not commissioned; externally peer reviewed.

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
