## [Reviewer comments · BMJ Open]

ARTICLE DETAILS

TITLE (PROVISIONAL)	Study protocol for 'The Project About Loneliness and Social networks (PALS)': a pragmatic, randomised trial comparing a facilitated social network intervention (Genie) with a wait-list control for lonely and socially-isolated people
AUTHORS	Band, Rebecca; Ewings, Sean; Cheetham-Blake, Tara; Ellis, Jaimie; Breheny, Katie; Vassilev, Ivaylo; Portillo, Mari Carmen; Yardley, Lucy; Blickem, C; Kandiyali, Rebecca; Culliford, David; Rogers, Anne

VERSION 1 – REVIEW

REVIEWER	Amelia Gulliver ANU, Australia
REVIEW RETURNED	25-Jan-2019

GENERAL COMMENTS	This paper presents a research protocol for the PALS (genie) trial. Overall, I found that much of the information I would expect to be in the protocol was missing (i.e., psychometric data and examples from each of the measures). Much of the information listed in the SPIRIT is not included in the main manuscript, and I would not expect to have to read through another document to find the information required. There is a lengthy document at the end that seems to be another version of the protocol, but it is not referred to in the manuscript, so I am unsure of why it has been included? This is the same for the other additional files – why are they not referred to in the main manuscript (i.e., file 1 presents the SPIRIT checklist). I would recommend revising this manuscript including all the information required by the SPIRIT checklist in to the main manuscript. Further minor comments are below. Page 6, L59 – an extra space prior to “facilitated”? Page 7, L40 “or in line with the usual working practices of the partner organisation” Can you provide an example of what this might be? Page 8, L6 – number and list criteria for ease of interpretation. Also, how will these criteria (e.g., “any condition which impacts upon ability, lack of capacity) be assessed and by whom? Page 8 – If the randomisation is computer generated, why are sequences being recorded at all “The sequences will be stored in sealed, opaque, numbered envelopes” – please clarify.
---

	Page 9, L 3 – state that written informed consent will be collected prior to the baseline – not just sought. Page 9 L6 – Only 15% dropout seems quite low for a trial of this nature – can you please confirm that the example trial you got this percentage from was similar – the reference is listed as: “Evaluation of the SF-12: usefulness of the mental health scale” – Although, this doesn’t seem to be the correct reference, please clarify? Page 12 – the outcomes are listed as “differences”, this seems unusual - I think you might mean just the concepts themselves – i.e., the primary outcome is health related quality of life or “mental health” as measured on the SF-12.
--	--

REVIEWER	Louise Hawkley NORC at the University of Chicago
REVIEW RETURNED	29-Jan-2019

GENERAL COMMENTS	Evaluations of the effectiveness of community-based interventions for loneliness and social isolation are very much needed. This paper describes a high quality protocol that will provide useful information about the feasibility, effectiveness, and cost-effectiveness of a specific type of intervention (Genie). This work could serve as a good model for others to evaluate their community-based efforts. It is valuable to see that this protocol describes an intervention that was intended to improve mental health writ large, but that is theoretically and practically well-aligned to address remediable risk factors for loneliness and social isolation. This should give heart to service agencies who “know” that their efforts are meaningful and effective in alleviating loneliness and isolation, even if that was not the intention. Hopefully seeing this protocol published will encourage other groups to conduct pragmatic randomized controlled trials. One overarching recommendation regarding the manuscript is to incorporate some of the “visual” aspects of the formal study protocol (i.e., p. 32 ff. in the submission) to help readers understand the design and flow of the study. For instance, the Study flow diagram (#3, p. 39 of 77) would provide a useful easy-to-follow graphic to understand the overall study design. Similarly, figure 10.5 in the protocol (p. 51) would be a helpful adjunct to the descriptions of recruitment and randomization in the manuscript (p. 8-9 of 77). Regarding outcomes, it would be helpful to define what is meant by “mental health” (the primary outcome) as measured using the SF-12. This occurs in several locations, and a description or definition on first appearance is recommended. In addition, please provide a bit more detail on (a) “participant engagement with new activities”: frequency of engagement? Number of new activities? Other?; (b) social network composition change: number of new members? new role types? Other?; and (c) health and social care use: measured how?
--

	The remaining comments are presented in roughly the order in which questions arose when reviewing the manuscript.  1. The Genie intervention is not described until p. 11, lines 13ff. A brief description would be valuable earlier in the manuscript (e.g., under Study Aims and Research Questions) and should include a repetition (from the title) of the meaning of the acronym (i.e., “Generating Engagement in Network Involvement”) to reinforce why this is called a “facilitated social network intervention.” If space permits, a brief explanation of Genie in the abstract would help browsing researchers to quickly identify potentially similar or relevant ideas and interventions. In addition, it would be helpful to include some detail about Genie later in the text; I’m curious whether the Genie resource database is maintained and regularly updated. Individuals who select organizations that have ceased to operate will be sorely disappointed and may lose interest and motivation to stick with the program. 2. Please identify the “three profiles of individuals who are ‘at-risk’” (p. 5, line 21) to help readers understand what sub-groups of the population you may be more likely to target and recruit for the trial. This will help readers understand who you’re referring to on p. 7, line 8, where you mention assessing PALS in “a community setting among at-risk populations.” 3. Some of the numbered cites in the introduction don’t seem to correspond with the references. For instance, supportive documentation for the link between loneliness and health service use (p. 5, lines 48-52) is only evident in ref #21, not refs #20 & 22. If the link exists through secondary sources (i.e., those cited by the authors of #20 and 22), I recommend citing the original sources. 4. The section labeled “Rationale and risk-benefits for the current trial” (p. 6) is lacking any information on risks or benefits. I’d be particularly interested in seeing some of the information you provide in the formal protocol regarding challenges for partner organizations (e.g., the last paragraph under ‘Ethical and Regulatory Considerations’, p. 64, lines 37-54). 5. Please define and describe ICECAP-A at first occurrence. 6. Please explain the purpose of the “perception of loneliness measure derived from the modified Brief Illness Perception questionnaire. I checked the source article and it’s still not clear what you’re trying to accomplish with this measure, and how it differs from the De Jong Gierveld Loneliness scale and the Campaign to End Loneliness scale. 7. Especially for those unfamiliar with the British health care system, explain how participants can report on health care, social service and other public sector costs (p. 17, lines 47-52). Do you
--	--

	mean participant usage of these services? Participant incurred costs, including the cost of formal carers, are more plausibly obtained directly from participants, but how well are participants able to remember and/or generate these cost estimates? Is there evidence that these questions have produced reliable information in past research? 8. What defines what is “optimal for society” (p. 17-18)? Is it solely cost? Are cost shifts to some sectors more valuable/optimal for society than other shifts? 9. Please define CUA (p. 18) at first appearance. 10. Sub-group analyses “will be planned prospectively” (p. 18, lines 15-17). Isn’t this manuscript the place to define the planned sub-group analyses? 11. Please develop/explain what is meant by the sentence that ends, “...considering how the underlying mechanisms and contexts relate to resource use and cost areas” (p. 18, lines 24-26). 12. The qualitative process refers to sampling participants “purposely based on circumstances of loneliness and sociodemographic factors” (p. 19, lines 20-22). Which sociodemographic factors and why?
--	--

VERSION 1 – AUTHOR RESPONSE

Reviewer 1

Comment	Response
Overall, I found that much of the information I would expect to be in the protocol was missing (i.e., psychometric data and examples from each of the measures).	We acknowledge that there is a lot of detailed information required in the protocol manuscript. However, in keeping with the author guidelines, which state that the manuscript should not exceed 4000 words, it is necessary to prioritise the information presented. We are using validated and widely used outcome measures, which are in the public domain. Therefore, we have not included information outlining the psychometric properties and examples from outcome measures in the text as this would involve a substantial increase in information presented (where there are several outcome measures). However, references are provided for these measures where relevant to direct the reader to this information.
Much of the information listed in the SPIRIT is not included in the main manuscript, and I would not expect to have to read through another document to find the information required. I would recommend revising this manuscript including all the information required by the SPIRIT checklist in to the main manuscript.	Of the 33 items within the SPIRIT checklist we have not included information relating to 25 items. Items we have amended/ included:  17a/ b – due to the nature of the study, it is not possible to blind anyone to participant allocation. However, the researchers conducting the baseline screening will be blinded to the randomisation sequence, and equally the statisticians will not be involved with the community partner and participant

	recruitment. We have added a brief sentence to Page 8 to clarify this. Several of these were not applicable or there was not sufficient space within the manuscript to add in items, especially when these relate specifically toward drug trials or non-pragmatic RCTs (as is the case in the current study). Please see updated checklist for further information.
There is a lengthy document at the end that seems to be another version of the protocol, but it is not referred to in the manuscript, so I am unsure of why it has been included? This is the same for the other additional files – why are they not referred to in the main manuscript (i.e., file 1 presents the SPIRIT checklist).	In the online system, there was the option to upload the study protocol (rather than the manuscript) for RCT protocol submissions; this is what this second document refers to: we will omit this from the manuscript revision to avoid confusion. Similarly, we uploaded the SPIRIT checklist as this is required by the online system. In the resubmitted manuscript, we have made the changes as advised by the editor.
Page 6, L59 – an extra space prior to “facilitated”?	Edited in manuscript version 2.
Page 7, L40 “or in line with the usual working practices of the partner organisation” Can you provide an example of what this might be?	The organisations we are working with are varied, and cover the spectrum of services, organisations and groups working within the community, and consequently so are our working practices. For example, this may relate to individuals who have been referred for befriending services, but may include those on waiting lists as well as newly referred participants. It may also include those identified in routine visits or may include those identified from screening lists of individuals in contact with a service or organisation.) We plan to publish more comprehensive information later in relation to our pre-implementation and implementation work, but have added a sentence to expand on this on page 7 of the revised manuscript.
Page 8, L6 – number and list criteria for ease of interpretation. Also, how will these criteria (e.g., “any condition which impacts upon ability, lack of capacity) be assessed and by whom?	We have included a sentence outlining this on page 8. Amended presentation of exclusion criteria in to a list in V2. Due to the pragmatic nature of the trial, all eligibility will be assessed by our community partners (i.e. there will be no formal assessment of ability or capacity) for those with the greatest need.
Page 8 – If the randomisation is computer generated, why are sequences being recorded at all “The sequences will be stored in sealed, opaque, numbered envelopes” – please clarify.	To clarify, the randomisation sequence was generated using a computer (using the “blockrand” package in R v3.5.1) as opposed to using a random number sequence in a book or some other process. This sequence is then implemented using envelopes (as the randomisation information is accessed following the baseline research visit – at which point participants are informed about their allocation).
Page 9, L 3 – state that written informed consent will be collected prior to the baseline – not just sought.	Modified in version 2 of the manuscript.
Page 9 L6 – Only 15% dropout seems quite low for a trial of this nature – can you please confirm that the example trial you got this percentage from was similar – the reference is listed as: “Evaluation of the SF-12: usefulness of the mental health scale” – Although, this doesn’t seem to be the correct reference, please clarify?	Apologies, this has been rectified now in the text the references have become misaligned in the preparation of the manuscript, thanks to the reviewer for pointing this out. This figure was based on estimates from a previous, similar trial; we have updated this to ensure the correct reference is provided. Although it is potentially fairly low, we will assess this (as well as other indicators such as

	follow-up) in relation to project progress and continuation as set out by the study funders (to be assessed in November 2019).
Page 12 – the outcomes are listed as “differences”, this seems unusual - I think you might mean just the concepts themselves – i.e., the primary outcome is health related quality of life or “mental health” as measured on the SF-12.	We agree with the reviewers comments, and as such, we have revised the manuscript to remove the word “differences” and instead, just listed the outcomes of interest (edits on page 13 of manuscript).

Reviewer 2

Comment	Response
One overarching recommendation regarding the manuscript is to incorporate some of the “visual” aspects of the formal study protocol (i.e., p. 32 ff. in the submission) to help readers understand the design and flow of the study. For instance, the Study flow diagram (#3, p. 39 of 77) would provide a useful easy-to-follow graphic to understand the overall study design. Similarly, figure 10.5 in the protocol (p. 51) would be a helpful adjunct to the descriptions of recruitment and randomization in the manuscript (p. 8-9 of 77).	We would like to thank the reviewer for this suggestion; we have included the study flow diagram (#3 in the main study protocol) as a Figure in the revised manuscript (added to the end of the manuscript). We have also added in the table to accompany the recruitment/ randomisation sections as suggested (Table 1, page 9).
Regarding outcomes, it would be helpful to define what is meant by “mental health” (the primary outcome) as measured using the SF-12. This occurs in several locations, and a description or definition on first appearance is recommended.	We have updated the text in the objectives to clarify that mental health refers to the SF-12 composite scale score. We have also added some additional text to explain what this subscale measures in the sample size consideration section on page 10.
In addition, please provide a bit more detail on (a) “participant engagement with new activities”: frequency of engagement? Number of new activities? Other?; (b) social network composition change: number of new members? new role types? Other?; and (c) health and social care use: measured how?	As a part of the Genie intervention participants map their network members according to three concentric circles of importance/ relevance. This includes type of relationships (e.g. partner, other family members, friends, acquaintances, hobby groups, activities), frequency of contact with each network member (e.g. every day, once a week, etc.), and their subjective importance (position in the network diagram from very important to least important). This data is collected at baseline and at 3-month follow up for intervention participants. This will allow us to explore changes in networks over time including overall number of network members and frequency of contact (Reeves et al. 2014), type of network (Vassilev et al. 2016), as well as number and types of new activities and relationships added at follow up, changes in the frequency of contact with and the subjective importance of existing network members (Vassilev et al. 2018). We have summarised this as “Social network composition change” within the manuscript.
The Genie intervention is not described until p. 11, lines 13ff. A brief description would be valuable earlier in the manuscript (e.g., under Study Aims and Research Questions) and should include a repetition (from the title) of the meaning of the acronym (i.e., “Generating Engagement	We have included some information about Genie previously located in the section on page 11 into the under ‘Study Aims and Research Questions’ section, as requested by the reviewer. Unfortunately, we do not have any additional space to add further description to the abstract.

in Network Involvement”) to reinforce why this is called a “facilitated social network intervention.” If space permits, a brief explanation of Genie in the abstract would help browsing researchers to quickly identify potentially similar or relevant ideas and interventions. In addition, it would be helpful to include some detail about Genie later in the text; I’m curious whether the Genie resource database is maintained and regularly updated. Individuals who select organizations that have ceased to operate will be sorely disappointed and may lose interest and motivation to stick with the program.	The database for this study is being managed and updated by the research team, and by community partners. We acknowledge that it may not be possible to ensure all inputs into the database are fully up to date – one advantage of working in collaboration with our community partners is that they are often those with the local knowledge about services and activities, and can bring this knowledge to the facilitation process.
Please identify the “three profiles of individuals who are ‘at-risk’” (p. 5, line 21) to help readers understand what sub-groups of the population you may be more likely to target and recruit for the trial. This will help readers understand who you’re referring to on p. 7, line 8, where you mention assessing PALS in “a community setting among at-risk populations.”	This refers to findings from a report published by the UK Office of National Statistics in 2018 which suggested there were three profiles of people who were likely to experience loneliness in the UK;  1/ older, female, homeowners who were likely widowed but better off financially 2/ middle-aged adults with long-term conditions, likely unemployed with poor physical health and economic status 3/ younger adults who were likely to be working and living with others but in areas they were not connected to (and which were more deprived). Our point in highlighting this was not to suggest that we will target these groups specifically – because we will not, it was simply to highlight that loneliness may be different and mean different things to different people, and for this reason we will be inclusive with participation in the current study. We have edited the text slightly on page 4 and also modified the text on page 5 – to replace “who are at risk of loneliness” to “who are experiencing loneliness” (which will be self-identified).
Some of the numbered cites in the introduction don’t seem to correspond with the references. For instance, supportive documentation for the link between loneliness and health service use (p. 5, lines 48-52) is only evident in ref #21, not refs #20 & 22. If the link exists through secondary sources (i.e., those cited by the authors of #20 and 22), I recommend citing the original sources	We apologise for this – as noted above, our references have become mismatched in the preparation of the manuscript. We have double-checked this in this revised version and hope this addresses the reviewers concern.
The section labeled “Rationale and risk-benefits for the current trial” (p. 6) is lacking any information on risks or benefits. I’d be particularly interested in seeing some of the information you provide in the formal protocol regarding challenges for partner organizations (e.g., the last paragraph under ‘Ethical and Regulatory Considerations’, p. 64, lines 37-54).	We agree these are important points, and would like to be able to outline them all in full, but due to space constraints we have added a brief summary of some of the key issues raised in the ‘Ethical and Regulatory Considerations’ section into the manuscript under the “Rationale and risk-benefits for the current trial” section.
Please define and describe ICECAP-A at first occurrence.	We have revised this throughout so that the measure is referred to as the "capability wellbeing measure" (ICECAP-A) throughout the manuscript,

	and have modified the outcome section to provide further information on the measure.
Please explain the purpose of the “perception of loneliness measure derived from the modified Brief Illness Perception questionnaire. I checked the source article and it’s still not clear what you’re trying to accomplish with this measure, and how it differs from the De Jong Gierveld Loneliness scale and the Campaign to End Loneliness scale.	We have included a modified version of the Brief illness perception questionnaire to assess cognitions regarding loneliness; we are interested in the extent to which cognitive-behavioural factors are important in the maintenance of loneliness, and to the best of our knowledge, no measures currently exist to capture this. However, the B-IPQ has been used widely among different patient groups and is able to be modified for specific conditions or symptoms (although we acknowledge this is not an illness nor a patient group). It assesses both cognitive and emotional representations – for example, that the symptom in question is likely to last a long time, or are within their personal control. We therefore hypothesize that this is different to the presence of loneliness as measured by the De Jong Gierveld or Campaign to End loneliness scale.
Especially for those unfamiliar with the British health care system, explain how participants can report on health care, social service and other public sector costs (p. 17, lines 47-52). Do you mean participant usage of these services? Participant incurred costs, including the cost of formal carers, are more plausibly obtained directly from participants, but how well are participants able to remember and/or generate these cost estimates? Is there evidence that these questions have produced reliable information in past research?	This is an excellent point and concern, which we agree with. Patient collected resource use instruments are not customarily subject to the same stringent validation process as other patient reported outcome measures. Work is ongoing see for example Ridyard, C.H. and D.A. Hughes, Methods for the collection of resource use data within clinical trials: a systematic review of studies funded by the UK health technology assessment program. Value in Health, 2010. 13(8): p. 867-872. We have also edited the use of “participant incurred costs” outlined in page 17 to “participant service use”.
What defines what is “optimal for society” (p. 17-18)? Is it solely cost? Are cost shifts to some sectors more valuable/optimal for society than other shifts?	The plan is to present decision makers' with information/evidence so that they can make the normative decision as to what is optimal. We as health economists do not prescribe what is optimal for society, and optimisation based on cost-effectiveness or cost-utility are normally limited to efficiency rather than need, equity, or fairness.
Please define CUA (p. 18) at first appearance.	We define CUA at first appearance on page 17; however, we have edited to use the full definition (cost-utility analysis) throughout the revised manuscript.
Sub-group analyses “will be planned prospectively” (p. 18, lines 15-17). Isn’t this manuscript the place to define the planned sub-group analyses?	Due to the constraints of the paper, we do not propose to report on any sub-group analyses here, other than to identify that this is likely to explore the targeting of the intervention (e.g. explaining different cost and effects in different groups of participants). This in part reflects our plan to work in a joined-up way with our process evaluation colleagues, as observations relating to the implementation of the intervention in different settings may inform the analyses we undertake. Nevertheless, the sub-group analyses will be pre-specified prior to starting the analysis in the combined statistical and health economic analyses plan.
Please develop/explain what is meant by the sentence that ends, “...considering how	We have referenced the MRC guidance (Moore, 2015) on process evaluations. When we talk about

the underlying mechanisms and contexts relate to resource use and cost areas” (p. 18, lines 24-26).	mechanisms of impact, we are considering how participants actively interact and respond to the intervention. However, we have edited this section within the revised manuscript to improve clarity. By context, we mean anything external to the intervention that may act as barrier or facilitator to the intervention. Working in a joined-up way with our process evaluation colleagues throughout will mean that we will be able to explain some of the likely differences in intervention costs and effects both within and across clusters.
The qualitative process refers to sampling participants “purposely based on circumstances of loneliness and sociodemographic factors” (p. 19, lines 20-22). Which sociodemographic factors and why?	We are sampling on several facets, in particular, this will include description of circumstances related to loneliness particularly nature of marginality from descriptions in trial recruitment (e.g. social anxiety, being new to a locality) as well as age, gender, locality, living arrangements or other factors (such as interpersonal status). We have added some more detail to reflect this in the text.